# Local Correlation among the Chiral Condensate, Monopoles, and Color Magnetic Fields in Abelian Projected QCD

**Hideo Suganuma** [1,*] and **Hiroki Ohata** [2]

1   Department of Physics, Kyoto University, Kitashirakawaoiwake, Sakyo, Kyoto 606-8502, Japan
2   Yukawa Institute for Theoretical Physics, Kyoto University, Sakyo, Kyoto 606-8502, Japan;
    hiroki.ohata@yukawa.kyoto-u.ac.jp
*   Correspondence: suganuma@scphys.kyoto-u.ac.jp

**Abstract:** Using the lattice gauge field theory, we study the relation among the local chiral condensate, monopoles, and color magnetic fields in quantum chromodynamics (QCD). First, we investigate idealized Abelian gauge systems of (1) a static monopole–antimonopole pair and (2) a magnetic flux without monopoles, on a four-dimensional Euclidean lattice. In these systems, we calculate the local chiral condensate on quasi-massless fermions coupled to the Abelian gauge field, and find that the chiral condensate is localized in the vicinity of the magnetic field. Second, using SU(3) lattice QCD Monte Carlo calculations, we investigate Abelian projected QCD in the maximally Abelian gauge, and find clear correlation of distribution similarity among the local chiral condensate, monopoles, and color magnetic fields in the Abelianized gauge configuration. As a statistical indicator, we measure the correlation coefficient $r$, and find a strong positive correlation of $r \simeq 0.8$ between the local chiral condensate and an Euclidean color-magnetic quantity $\mathcal{F}$ in Abelian projected QCD. The correlation is also investigated for the deconfined phase in thermal QCD. As an interesting conjecture, like magnetic catalysis, the chiral condensate is locally enhanced by the strong color-magnetic field around the monopoles in QCD.

**Keywords:** QCD; chiral symmetry; monopole; lattice QCD; spontaneous symmetry breaking; Abelian projection; magnetic catalysis





## 1. Introduction

Quantum chromodynamics (QCD) is an SU($N_c$) gauge theory to describe the strong interaction, and has presented many interesting subjects full of variety and difficult problems in physics. Actually, in spite of the simple form of the QCD action, this miracle theory creates hundreds of hadrons and leads to various interesting non-perturbative phenomena, such as color confinement and dynamical chiral-symmetry breaking [1].

This magic is due to the strong coupling of QCD in the low-energy region, and this strong-coupling nature drastically changes the vacuum structure itself. Therefore, a perturbative technique is no more workable and analytical treatment of QCD is fairly difficult in the strong-coupling region. As a reliable standard technique, lattice QCD Monte Carlo simulations have been applied to analyze non-perturbative QCD [2,3].

Among the non-perturbative properties of QCD, spontaneous chiral-symmetry breaking is particularly important in our real world. Indeed, chiral symmetry breaking drastically influences the vacuum structure and gives a non-trivial vacuum expectation value of the chiral condensate $\langle \bar{q}q \rangle$, which plays the role of an order parameter. Additionally, it is considered that chiral symmetry breaking leads to dynamical quark-mass generation [1,4], and creates most of the matter mass of our Universe, apart from the dark matter, because only small masses of u, d, current quarks, and electrons are Higgs-origin in atoms [5] and their contribution to the nucleon mass is estimated to be small [6]. In addition, chiral symmetry breaking inevitably accompanies light pions of the Nambu–Goldstone bosons, and their small mass gives range of the nuclear force.

In non-perturbative QCD, color confinement is also one of the most important phenomena in physics, and presents an extremely difficult mathematical problem. Experiments for hadron spectra and lattice QCD studies for various inter-quark potentials [7–10] show that the quark confining force is basically characterized by a universal physical quantity of the string tension $\sigma \simeq 0.89$ GeV/fm. This universal string tension is physically explained by one-dimensional squeezing of the color electric flux, i.e., the color flux-tube formation in hadrons, as is also indicated by lattice QCD for both mesons [3] and baryons [11]. As for the relation between color confinement and chiral symmetry breaking, it is not yet clarified directly from QCD. Although almost coincidence between deconfinement and chiral-restoration temperatures [12] suggests their close correlation, a lattice QCD analysis using the Dirac-mode expansion based on the Banks–Casher relation [13] indicates some independence of these phenomena in QCD [14,15].

For the quark confinement mechanism, Nambu [16], 't Hooft [17], and Mandelstam [18] proposed the dual superconductivity scenario, paying attention to analogy with the Abrikosov vortex in the superconductivity, where Cooper-pair condensation leads to the Meissner effect, and the magnetic flux is excluded or squeezed like a one-dimensional tube as the Abrikosov vortex. If the QCD vacuum can be regarded as the dual version of the superconductor, the electric-type color flux is squeezed between (anti)quarks in hadrons, and quark confinement can be physically explained by the dual Meissner effect. Because of the electromagnetic duality, the dual Meissner effect inevitably needs condensation of magnetic objects, i.e., color magnetic monopoles, which correspond to the dual version of the electric Cooper-pair bosonic field.

In the dual-superconductor picture for the QCD vacuum, however, there are two large gaps with QCD.

1. Although QCD is a non-Abelian gauge theory, the dual-superconductor picture is based on an Abelian gauge theory subject to the Maxwell-type equations including magnetic currents, where electromagnetic duality is manifest;
2. Although QCD includes only color electric variables, i.e., quarks and gluons, as the elementary degrees of freedom, the dual-superconductor picture requires condensation of color magnetic monopoles as a key concept.

Historically, to bridge between QCD and the dual-superconductivity, 't Hooft proposed Abelian gauge fixing [19], partial gauge fixing which only remains Abelian gauge degrees of freedom in QCD. By Abelian gauge fixing, QCD reduces into an Abelian gauge theory, where off-diagonal gluons behave as charged matter fields similar to $W_\mu^\pm$ in the Weinberg–Salam model and give the color electric current $j_\mu$ in terms of the residual Abelian gauge symmetry. As a remarkable fact in the Abelian gauge, color-magnetic monopoles appear as topological objects corresponding to the non-trivial homotopy group $\Pi_2(\mathrm{SU}(N_c)/\mathrm{U}(1)^{N_c-1}) = \mathbf{Z}_\infty^{N_c-1}$ in a similar manner to appearance of 't Hooft–Polyakov monopoles [20] in the SU(2) non-Abelian Higgs theory. Thus, in the Abelian gauge, QCD is reduced into an Abelian gauge theory, including both electric current $j_\mu$ and magnetic current $k_\mu$, which is expected to give a theoretical basis of the dual-superconductor picture for the confinement mechanism, although off-diagonal gluons remain as charged matter fields.

From the viewpoint of Abelianzation of QCD, the maximally Abelian (MA) gauge [21] is an interesting special Abelian gauge. In the MA gauge, off-diagonal gluons have a large effective mass of about 1 GeV in both SU(2) and SU(3) lattice QCD [22–24], so that off-diagonal gluons become infrared inactive, and only the Abelian gluon is relevant at distances larger than about 0.2 fm. Additionally, monopole condensation is suggested from appearance of long entangled monopole worldlines [21,25] and the magnetic screening in lattice QCD [26,27].

In this way, by taking the MA gauge, the QCD vacuum can be regarded as an Abelian dual superconductor at a large scale, and color magnetic monopoles seem to capture essence of non-perturbative QCD. Note, however, that, even without gauge fixing, there is an evidence of monopole condensation in non-Abelian gauge theories [27], and, therefore, it might be possible to define infrared-relevant monopoles in QCD and to construct the dual

superconductor system in more general manner. In fact, MA gauge fixing gives a concrete way to extract infrared-relevant Abelian gauge manifold and monopoles from QCD.

In the context of the dual superconductor picture, close correlation between monopoles and chiral symmetry breaking was pointed out in the dual Ginzburg–Landau theory [28], in SU(2) lattice QCD in the MA gauge [29,30], and in SU(3) lattice QCD [31,32]. Since most of the pioneering lattice studies were done in SU(2) QCD or on a small lattice as $8^3 \times 4$, we recently investigated SU(3) QCD with a large volume, and find a clear correlation between monopoles and the chiral condensate in SU(3) lattice QCD in the MA gauge [33].

In this paper, as a continuation of Ref. [33], we proceed the lattice works for the relation between chiral symmetry breaking and color magnetic objects including monopoles. In particular, as a new point of this paper, we quantitatively study correlation of the local chiral condensate with color magnetic fields using the lattice gauge theory.

The organization of this paper is as follows. In Section 2, we review the MA gauge and Abelianization of QCD in SU(3) lattice formalism. In Section 3, we prepare magnetic objects in Abelian projected QCD. In Section 4, we consider the local chiral condensate and chiral symmetry breaking in Abelian gauge systems. In Section 5, we present idealized Abelian gauge systems of a static monopole–antimonopole pair on a lattice, and investigate the relation of the local chiral condensate with the magnetic objects. In Section 6, we perform SU(3) lattice QCD Monte Carlo calculations and study the relation among monopoles, magnetic fields, and the local chiral condensate in Abelian projected QCD in the MA gauge. Section 7 is devoted for summary and conclusion.

## 2. Maximally Abelian Gauge and Abelianization of QCD

To begin with, we briefly review the lattice formalism for maximally Abelian (MA) gauge fixing and Abelianization in QCD.

Continuum QCD is described with the quark field $q(x)$, the gluon field $A_\mu(x) \in \mathrm{su}(N_c)$ and the QCD gauge coupling $g$. In SU($N_c$) lattice QCD [3], the gluon field is described as the SU($N_c$) link variable $U_\mu(s) \equiv \exp(iag A_\mu(s)) \in$ SU($N_c$) on four-dimensional Euclidean lattices with the spacing $a$ and the volume $V = L_x L_y L_z L_t$.

Using the Cartan subalgebra $\vec{H} \equiv (T_3, T_8)$ in SU(3), MA gauge fixing is defined so as to maximize

$$R_{\mathrm{MA}}[U_\mu(s)] \equiv \sum_s \sum_{\mu=1}^4 \mathrm{tr}\left( U_\mu^\dagger(s) \vec{H} U_\mu(s) \vec{H} \right) = \sum_s \sum_{\mu=1}^4 \left( 1 - \frac{1}{2} \sum_{i \neq j} |U_\mu(s)_{ij}|^2 \right) \tag{1}$$

by the SU(3) gauge transformation, and, therefore, this gauge fixing strongly suppresses all the off-diagonal fluctuation of the SU(3) gauge field. In the MA gauge, the SU(3) gauge group is partially fixed remaining its maximal torus subgroup U(1)$_3$ × U(1)$_8$ with the global Weyl (color permutation) symmetry [34], and QCD is reduced to an Abelian gauge theory.

From the SU(3) link variable $U_\mu^{\mathrm{MA}}(s) \in$ SU(3) in the MA gauge, we extract the Abelian link variable

$$u_\mu(s) = e^{i\vec{\theta}_\mu(s) \cdot \vec{H}} = \mathrm{diag}\left( e^{i\theta_\mu^1(s)}, e^{i\theta_\mu^2(s)}, e^{i\theta_\mu^3(s)} \right) \in \mathrm{U(1)}_3 \times \mathrm{U(1)}_8 \subset \mathrm{SU(3)} \tag{2}$$

by maximizing the overlap

$$R_{\mathrm{Abel}} \equiv \frac{1}{3} \mathrm{Re}\,\mathrm{tr}\left\{ U_\mu^{\mathrm{MA}}(s) u_\mu^\dagger(s) \right\} \in \left[ -\frac{1}{2}, 1 \right]. \tag{3}$$

Note that the distance between $u_\mu(s)$ and $U_\mu^{\mathrm{MA}}(s)$ becomes the smallest in the SU(3) manifold, and there is a constraint $\sum_{i=1}^3 \theta_\mu^i(s) = 0 \pmod{2\pi}$ reflecting the uni-determinant of $u_\mu(s)$. Here, $\theta_\mu^i(s)$ ($i = 1, 2, 3$) is taken to be the principal value of $-\pi \leq \theta_\mu^i(s) < \pi$.

The Abelian projection is defined by the simple replacement of the SU(3) link variable $U_\mu(s)$ by the Abelian link variable $u_\mu(s)$ for each gauge configuration, that is, $O[U_\mu(s)] \rightarrow$

$O[u_\mu(s)]$ for QCD operators. Abelian projected QCD is thus extracted from SU(3) QCD. The case of $\langle O[U_\mu(s)]\rangle \simeq \langle O[u_\mu(s)]\rangle$ is called "Abelian dominance" for the operator $O$ [35].

As a remarkable fact, Abelian dominance of quark confinement is shown in both SU(2) [36] and SU(3) lattice QCD [37–39]. Additionally, Abelian dominance of chiral symmetry breaking is observed in SU(2) [29–31] and SU(3) lattice QCD [33].

## 3. Magnetic Objects in Abelian Projected QCD

In this section, we prepare magnetic objects in Abelian projected QCD in four-dimensional Euclidean space-time.

### 3.1. Monopoles in Abelian Projected QCD

In this subsection, we define monopoles in Abelian projected QCD in lattice formalism [40]. Like the ordinary SU(3) plaquette, the Abelian plaquette variable is defined as

$$
\begin{aligned}
u_{\mu\nu}(s) &\equiv u_\mu(s)u_\nu(s+\hat{\mu})u_\mu^\dagger(s+\hat{\nu})u_\nu^\dagger(s) = e^{i\vec{\theta}_{\mu\nu}(s)\cdot\vec{H}} \\
&= \mathrm{diag}(e^{i\theta_{\mu\nu}^1(s)}, e^{i\theta_{\mu\nu}^2(s)}, e^{i\theta_{\mu\nu}^3(s)}) \in \mathrm{U}(1)^2 \subset \mathrm{SU}(3),
\end{aligned}
\tag{4}
$$

where $\hat{\mu}$ is the $\mu$-directed unit vector in the lattice unit. The Abelian field strength $\theta_{\mu\nu}^i(s)$ ($i = 1, 2, 3$) is the principal value of the exponent in $u_{\mu\nu}(s)$, and is defined as

$$
\begin{aligned}
\partial_\mu\theta_\nu^i(s) - \partial_\nu\theta_\mu^i(s) &= \theta_{\mu\nu}^i(s) - 2\pi n_{\mu\nu}^i(s), \\
-\pi \leq \theta_{\mu\nu}^i(s) &< \pi, \quad n_{\mu\nu}^i(s) \in \mathbb{Z},
\end{aligned}
\tag{5}
$$

with the forward derivative $\partial_\mu$. Note that $\theta_{\mu\nu}^i(s)$ is $\mathrm{U}(1)^2$ gauge invariant and corresponds to the regular Abelian field strength in the continuum limit of $a \to 0$, while $n_{\mu\nu}^i(s)$ corresponds to the singular gauge-variant Dirac string [40].

The electric current $j_\mu^i$ and the monopole current $k_\mu^i$ are defined from the Abelian field strength $\theta_{\mu\nu}^i$ as

$$
\begin{aligned}
j_\nu^i(s) &\equiv \partial_\mu'\theta_{\mu\nu}^i(s), \\
\end{aligned}
\tag{6}
$$

$$
\begin{aligned}
k_\nu^i(s) &\equiv \partial_\mu\tilde{\theta}_{\mu\nu}^i(s)/2\pi = \partial_\mu\tilde{n}_{\mu\nu}^i(s) \in \mathbb{Z},
\end{aligned}
\tag{7}
$$

where $\partial_\mu'$ is the backward derivative and $\tilde{\theta}_{\mu\nu}$ is the dual tensor of $\tilde{\theta}_{\mu\nu} \equiv \frac{1}{2}\epsilon_{\mu\nu\alpha\beta}\theta_{\alpha\beta}$. Both electric and monopole currents are $\mathrm{U}(1)^2$ gauge invariant, according to $\mathrm{U}(1)^2$ gauge invariance of $\theta_{\mu\nu}^i(s)$. In the lattice formalism, $k_\mu^i(s)$ is located at the dual lattice $L_{\mathrm{dual}}^4$ of $s^\alpha + \frac{1}{2}$ with flowing in $\mu$ direction [41].

In this way, Abelian projected QCD includes both electric current $j_\mu^i$ and monopole current $k_\mu^i$. Remarkably, lattice QCD shows monopole dominance, i.e., dominant role of monopoles for quark confinement in the MA gauge [42]. Additionally, lattice QCD shows monopole dominance for chiral symmetry breaking, that is, monopoles in the MA gauge crucially contribute to spontaneous chiral-symmetry breaking in both SU(2) [29,31] and SU(3) lattice QCD [33].

In the lattice formalism, the monopole current $k_\mu^i$ appears on the dual lattice $L_{\mathrm{dual}}^4$ of $s^\alpha + \frac{1}{2}$, and, therefore, we define the local monopole density

$$
\rho_{\mathrm{L}}(s) \equiv \frac{1}{3 \cdot 2^4} \sum_{i=1}^3 \sum_{s' \in P(s)} \sum_{\mu=1}^4 \left|k_\mu^i(s')\right|,
\tag{8}
$$

where $P(s)$ denotes the dual lattices in the vicinity of $s$, i.e., $P(s) = \left\{s' \in L_{\mathrm{dual}}^4 \big| |s - s'| = 1\right\}$. Note here that the distance between the site $s$ and its closest dual site $s'$ is $|s - s'| = \sqrt{\sum_1^4(\frac{1}{2})^2} = 1$ in the four-dimensional Euclidean space-time.

### 3.2. General Argument for Magnetic Instability and Magnetic Objects in the QCD Vacuum

In QCD in the MA gauge, color magnetic monopoles generally appear, and play an important role in non-perturbative properties, which might looks curious, since the original QCD action does not have monopoles.

However, some active roles of magnetic objects would be natural in QCD, because QCD itself has color magnetic instability, and spontaneous generation of color magnetic fields generally takes place, as Savvidy first pointed out in 1977 [43,44].

In fact, in the QCD vacuum in the Minkowski space-time, the gluon condensate $\langle G_{\mu\nu}^a G_{\mu\nu}^a \rangle$ takes a large positive value, which physically means that the QCD vacuum is filled with color magnetic fields. Since the gluon condensate is expressed with color magnetic fields $\vec{H}_a$ and color electric fields $\vec{E}_a$ as

$$\langle G_{\mu\nu}^a G_{\mu\nu}^a \rangle = 2(\langle \vec{H}_a^2 \rangle - \langle \vec{E}_a^2 \rangle) > 0, \tag{9}$$

its large positivity means inevitable significant generation of color magnetic fields. Thus, some superior role of magnetic objects is expected instead of electric objects in the Minkowski QCD vacuum.

In the Euclidean space-time, because of the space-time $SO(4)$ symmetry, the roles of magnetic and electric fields become similar. Actually, the gluon condensate is written as $G_{\mu\nu}^a G_{\mu\nu}^a = 2(\vec{H}_a^2 + \vec{E}_a^2)$, where the electromagnetic duality is manifest. Then, in Euclidean QCD, the electric field often behaves as a magnetic field, and, therefore, we regard the Euclidean electric field as a sort of the magnetic field in this paper.

### 3.3. Lorentz Invariant Quantities in Abelian Projected QCD

In Abelian projected QCD, there are two Lorentz invariant quantities $\mathcal{F}$ and $\mathcal{G}$ in the Euclidean space-time:

$$\mathcal{F} \equiv \frac{1}{3} \sum_{i=1}^{3} \frac{1}{4} F_i^{\mu\nu} F_i^{\mu\nu} = \frac{1}{3} \sum_{i=1}^{3} \frac{1}{2} (\vec{H}_i^2 + \vec{E}_i^2), \tag{10}$$

$$\mathcal{G} \equiv \frac{1}{3} \sum_{i=1}^{3} \frac{1}{4} F_i^{\mu\nu} \tilde{F}_i^{\mu\nu} = \frac{1}{3} \sum_{i=1}^{3} \vec{H}_i \cdot \vec{E}_i, \tag{11}$$

with the color magnetic field $(\vec{H}_i)_j \equiv \frac{1}{2} \epsilon_{jkl} F_i^{kl}$ and the color electric field $(\vec{E}_i)_j \equiv F_i^{j4}$. These quantities are also invariant under the residual U(1)$^2$ gauge transformation and global Weyl transformation [34], i.e., permutation of the color index, in the MA gauge.

Here, $\mathcal{F}$ is parity-even and expresses total magnitude of magnetic fields in the Euclidean space-time, since the electric field behaves as a magnetic field there. In this paper, we simply call $\mathcal{F}$ "magnetic quantity" in Euclidean gauge theories. Note that $\mathcal{G}$ is parity-odd and is just the Abelian projected quantity of the topological charge density on instantons in QCD, which might relate to chiral symmetry breaking.

In the lattice formalism, the field strength tensor is a plaquette variable spanning at $s$, $s + \hat{\mu}$, $s + \hat{\nu}$, and $s + \hat{\mu} + \hat{\nu}$, so that we define the Abelian field strength $F_{\mu\nu}^i(s)$ as the local average of clover-type four plaquettes,

$$a^2 g F_{\mu\nu}^i(s) \equiv \frac{1}{4} \left( \theta_{\mu\nu}^i(s) + \theta_{\mu\nu}^i(s + \hat{\mu}) + \theta_{\mu\nu}^i(s + \hat{\nu}) + \theta_{\mu\nu}^i(s + \hat{\mu} + \hat{\nu}) \right), \tag{12}$$

and consider $\mathcal{F}$ and $\mathcal{G}$ as local quantities in each Abelian gauge configuration.

## 4. Local Chiral Condensate and Chiral Symmetry Breaking in Gauge Theories

In this section, we consider the chiral condensate and chiral symmetry breaking in the gauge theory in terms of the quark propagator.

### 4.1. Local Chiral Condensate in Lattice QCD

In this subsection, we briefly review the local chiral condensate in lattice formalism. The local chiral condensate can be calculated with the quark propagator for each gauge configuration $U = \{U_\mu(s)\}$ generated with the Monte Carlo method.

As the lattice fermion, we here adopt the Kogut–Susskind (KS) fermion [3]. For the KS fermion, the Dirac operator $\gamma_\mu D_\mu$ is expressed by $\eta_\mu D_\mu$ with the staggered phase $\eta_\mu(s) \equiv (-1)^{s_1 + \cdots + s_{\mu-1}}$ ($\mu \geq 2$) with $\eta_1(s) \equiv 1$. The KS Dirac operator is expressed as

$$(\eta_\mu D_\mu)_{ss'} = \frac{1}{2} \sum_{\mu=1}^{4} \sum_{\pm} \pm \eta_\mu(s) U_{\pm\mu}(s) \delta_{s \pm \hat\mu, s'} \tag{13}$$

with $U_{-\mu}(s) \equiv U_\mu^\dagger(s - \hat\mu)$, and the KS Dirac eigenvalue equation takes the form of

$$\frac{1}{2} \sum_{\mu=1}^{4} \sum_{\pm} \pm \eta_\mu(s) U_{\pm\mu}(s) \chi_n(s \pm \hat\mu) = i\lambda_n \chi_n(s). \tag{14}$$

Here, the quark field $q^\alpha(s)$ is described by a spinless Grassmann variable $\chi(s)$ [3], and the chiral condensate per flavor is evaluated as $\langle \bar q q \rangle = \langle \bar\chi\chi \rangle / 4$ in the continuum limit.

The local chiral condensate can be calculated using the quark propagator of the KS fermion with a small quark mass $m$. The chiral-limit value is estimated by the chiral extrapolation of $m \to 0$. As a technical caution, the chiral and continuum limits do not commute for the KS fermion at the quenched level, although this problem would be absent in full QCD [45].

For the gauge configuration $U = \{U_\mu(s)\}$, the Euclidean KS fermion propagator is given by

$$G_U^{ij}(x, y) \equiv \langle \chi^i(x) \bar\chi^j(y) \rangle_U = \langle x, i | \left( \frac{1}{\eta_\mu D_\mu[U] + m} \right) | y, j \rangle \tag{15}$$

with the color index $i$ and $j$. This propagator is numerically obtained by solving the large-scale linear equation with a point source. The local chiral condensate for the gauge configuration $\{U_\mu(s)\}$ is expressed with the propagator as

$$\langle \bar\chi(x)\chi(x) \rangle_U = -\operatorname{Tr} G_U(x, x). \tag{16}$$

Here, we consider the net chiral condensate by subtracting the contribution from the trivial vacuum $U = 1$ as

$$\langle \bar\chi\chi(x) \rangle_U \equiv \langle \bar\chi(x)\chi(x) \rangle_U - \langle \bar\chi\chi \rangle_{U=1}, \tag{17}$$

where the subtraction term is exactly zero in the chiral limit $m = 0$. The global chiral condensate is obtained by taking its average over the space-time $x$ and the gauge ensembles $U_1, U_2, ..., U_N$,

$$\langle \bar\chi\chi \rangle \equiv \sum_{x,i} \langle \bar\chi\chi(x) \rangle_{U_i} / \sum_{x,i} 1. \tag{18}$$

### 4.2. Chiral Symmetry Breaking in Abelian Gauge System

In this subsection, we analytically investigate relation between chiral symmetry breaking and the field strength in Euclidean Abelian gauge systems. For the simple argument, we consider Euclidean U(1) gauge systems with quasi-massless Dirac fermions coupled to the U(1) gauge field, although it is straightforward to generalize this argument to Abelian projected QCD with U(1)$^2$ gauge symmetry.

In the U(1) gauge system, the chiral condensate is proportional to the functional trace of the fermion propagator,

$$I \equiv \mathrm{Tr}\frac{1}{\slashed{D}+m} = -m\mathrm{Tr}\frac{1}{D^2 - m^2 + \frac{g}{2}\sigma \cdot F},$$ (19)

with the covariant derivative $D_\mu \equiv \partial_\mu + igA_\mu$, the field strength $F_{\mu\nu}$, $\sigma \cdot F \equiv \sigma_{\mu\nu}F_{\mu\nu}$ and $\sigma_{\mu\nu} \equiv \frac{i}{2}[\gamma_\mu, \gamma_\nu]$. Note that $D^2 - m^2$ is a negative-definite operator, and all of its eigenvalues are negative. Since the trace of any odd-number product of $\gamma$-matrices is zero, we find

$$I = -m\mathrm{Tr}\frac{D^2 - m^2 - \frac{g}{2}\sigma \cdot F}{(D^2 - m^2)^2 - 2g^2(\mathcal{F} - \gamma_5\mathcal{G}) - \frac{g}{2}[D^2, \sigma \cdot F]},$$ (20)

with

$$\mathcal{F} \equiv \frac{1}{4}F_{\mu\nu}F_{\mu\nu} = \frac{1}{2}(\vec{H}^2 + \vec{E}^2), \quad \mathcal{G} \equiv \frac{1}{4}F_{\mu\nu}\tilde{F}_{\mu\nu} = \vec{H} \cdot \vec{E}.$$ (21)

Because of the overall factor $m$ in $I$, $I$ goes to zero in the chiral limit $m \to 0$, unless the denominator becomes zero in this limit.

Since the operator $(D^2 - m^2)^2$ in the denominator is positive definite, to realize the zero denominator in $I$ in Equation (20), we need a significant negative contribution from the other three terms including $\mathcal{F}$, $\mathcal{G}$, or $[D^2, \sigma \cdot F]$. For instance, in the absence of the field strength, i.e., $F_{\mu\nu} \equiv 0$, one finds near $m \simeq 0$

$$I_{F_{\mu\nu}\equiv 0} = m\mathrm{Tr}\frac{1}{p^2 + m^2} = m\gamma \int \frac{d^4p}{(2\pi)^4}\frac{1}{p^2 + m^2} = m\frac{\gamma}{16\pi^2}\int^{\Lambda^2} dp^2 \frac{p^2}{p^2 + m^2} \simeq m\frac{\gamma\Lambda^2}{16\pi^2}$$ (22)

with the UV cut-off $\Lambda$ and the degeneracy $\gamma$. According to the positive denominator in the integrand, $I_{F_{\mu\nu}\equiv 0}$ has no IR singularity to cancel $m$ of the numerator, and, therefore, $I_{F_{\mu\nu}\equiv 0}$ goes to zero in the chiral limit of $m \to 0$.

To cancel $m$ in the numerator of $I$, we need a significantly large amount of the field strength so as to present zero mode in the denominator of $I$ and to keep $I$ non-zero in the chiral limit. Note here that $\mathcal{F}(\geq 0)$ always gives a negative (non-positive) contribution in the denominator of $I$, while the contribution from $\mathcal{G}$ or $[D^2, \sigma \cdot F]$ can be positive and negative. In fact, the magnetic quantity $\mathcal{F}$ can give the zero mode in the denominator of $I$, even without the contribution from $\mathcal{G}$ and $[D^2, \sigma \cdot F]$. In contrast, in Euclidean Abelian gauge systems, $\mathcal{F} \equiv \frac{1}{4}F_{\mu\nu}^2 = 0$ means $F_{\mu\nu} = 0$, and then $\mathcal{G} = [D^2, \sigma \cdot F] = 0$.

To conclude, the magnetic quantity $\mathcal{F}$ is expected to be significantly important to realize chiral symmetry breaking in Euclidean Abelian gauge theories, although, in some cases, the contribution from $\mathcal{G}$ and $[D^2, \sigma \cdot F]$ can assist the realization of chiral symmetry breaking.

In a special case of constant $F_{\mu\nu}$, one finds $[D^2, \sigma \cdot F] = 0$ for the Abelian system, and obtains

$$I = -m\mathrm{Tr}\frac{(D^2 - m^2)[(D^2 - m^2)^2 - 2g^2\mathcal{F}]}{[(D^2 - m^2)^2 - 2g^2\mathcal{F}]^2 - 4g^2\mathcal{G}^2},$$ (23)

because of $\mathrm{tr}\gamma_5 = \mathrm{tr}\sigma_{\mu\nu} = \mathrm{tr}\gamma_5\sigma_{\mu\nu} = 0$. For more special case of a constant magnetic field, there occurs the Landau-level quantization, and the spatial degrees of freedom perpendicular to the magnetic field is frozen in the lowest Landau level. This infrared effective low-dimensionalization of the charged spinor dynamics induces chiral symmetry breaking in the chiral limit [46–48], which is known as magnetic catalysis [49].

## 5. Abelian Gauge System with a Static Monopole–Antimonopole Pair on a Lattice

In the QCD vacuum, complicated monopole world-lines generally emerge in the MA gauge [21,25], and, therefore, it is difficult to clarify the primary correlation with the chiral condensate among the magnetic objects, such as monopoles, $\mathcal{F}$ and $\mathcal{G}$.

In this section, to seek for the primary correlation with the chiral condensate, we create idealized Abelian gauge system with a monopole–antimonopole pair on a lattice, and investigate the relation among the local chiral condensate, monoples, and magnetic fields. Additionally, we consider a magnetic flux system without monopoles.

For simplicity, we here consider U(1) lattice gauge systems described by U(1) link variables

$$u_\mu(s) = e^{i\theta_\mu(s)} \in \mathrm{U}(1), \tag{24}$$

and quasi-massless Dirac fermions coupled to U(1) gauge fields with the coupling $g = 1$.

### 5.1. Static Monopole–Antimonopole Pair Systems

To begin with, we deal with an idealized Abelian gauge system of a static monopole–antimonopole pair on a periodic lattice of the four-dimensional Euclidean space-time.

In the three-dimensional space $\mathbf{R}^3$, let us consider a static monopole–antimonopole pair with the distance of $l$ in $z$-direction. To realize such a lattice gauge system, we set the Abelian link-variable $u_\mu(s)$ to be

$$u_x(s) = u_y(s+\hat{x}) = u_x^\dagger(s+\hat{y}) = u_y^\dagger(s) = i \quad \text{for} \quad s_x = s_y = 0, \ 1 \le s_z \le l, \tag{25}$$

otherwise $u_\mu(s) = 1$.

Figure 1 shows the building-block plaquette to realize a static monopole–antimonopole pair on the lattice. Here, only the red link-variables take a non-trivial value of $i$.

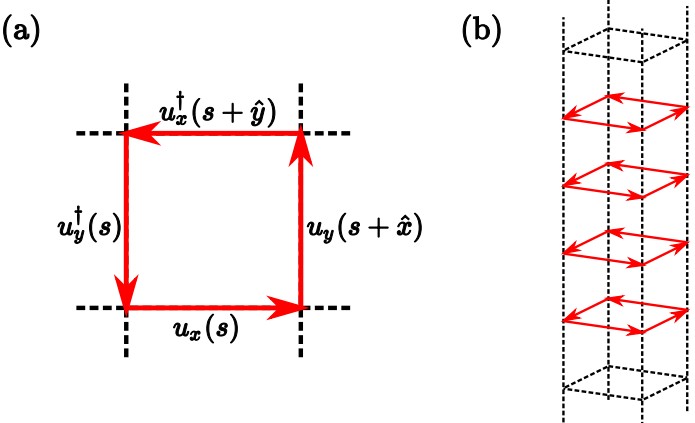

**Figure 1.** The building-block plaquette to realize a static monopole–antimonopole pair on the lattice in (**a**) the $x$-$y$ plane and (**b**) spatial $\mathbf{R}^3$ for $s = (0, 0, s_z, s_t)$ with $1 \le s_z \le l$. Only the red link-variables take a non-trivial value of $i$. The all-red plaquette induces the singular Dirac string at its center on the dual lattice. A physical magnetic field is also created in the neighboring plaquette $u_{xy}(s)$ including only one red link.

As for the phase variable $\theta_\mu(s)$, which corresponds to the Abelian gluon, one finds

$$\theta_x(s) = \theta_y(s+\hat{x}) = -\theta_x(s+\hat{y}) = -\theta_y(s) = \frac{\pi}{2} \quad \text{for} \quad s_x = s_y = 0, \ 1 \le s_z \le l, \tag{26}$$

otherwise $\theta_\mu(s) = 0$. For the all-red plaquette with $s_x = s_y = 0$ and $1 \le s_z \le l$, one gets

$$\partial_x \theta_y(s) - \partial_y \theta_x(s) = \theta_x(s) + \theta_y(s+\hat{x}) - \theta_x(s+\hat{y}) - \theta_y(s) = 2\pi, \tag{27}$$

which leads to the Dirac string of $n_{xy}(s) = -1$ and zero field strength $\theta_{xy}(s) = 0$, because of the definition of the field strength $\theta_{\mu\nu}(s)$ and the Dirac string $n_{\mu\nu}(s)$,

$$\partial_\mu \theta_\nu(s) - \partial_\nu \theta_\mu(s) = \theta_{\mu\nu}(s) - 2\pi n_{\mu\nu}(s), \quad -\pi \leq \theta_{\mu\nu}(s) < \pi, \quad n_{\mu\nu}(s) \in \mathbb{Z}. \quad (28)$$

Thus, the all-red plaquette induces the singular Dirac string at its center on the dual lattice. In fact, for the idealized system in Figure 1b, a Dirac string appears inside the all-red plaquette.

At the terminal of the Dirac string, a monopole or an anti-monopole appears on the dual lattice, as shown in Figure 2. Actually, the three-dimensional spatial cube including only one all-red plaquette has a static (anti)monopole at its center (on the dual lattice), because only one $n_{kl}(s)$ has non-zero value of $\pm 1$ among the six independent plaquettes composing the cube,

$$\begin{aligned} k_4(s) &= \partial_j \tilde{n}_{j4}(s) = \frac{1}{2}\epsilon_{jkl}\partial_j n_{kl}(s) = \frac{1}{2}\epsilon_{jkl}\{n_{kl}(s+\hat{j}) - n_{kl}(s)\} \\ &= n_{xy}(s+\hat{z}) - n_{xy}(s) + n_{yz}(s+\hat{x}) - n_{yz}(s) + n_{zx}(s+\hat{y}) - n_{zx}(s) \\ &= \pm 1. \end{aligned} \quad (29)$$

Thus, this idealized system includes a static monopole at $(\frac{1}{2}, \frac{1}{2}, \frac{1}{2})$ and a static anti-monopole at $(\frac{1}{2}, \frac{1}{2}, l + \frac{1}{2})$ in spatial $\mathbf{R}^3$.

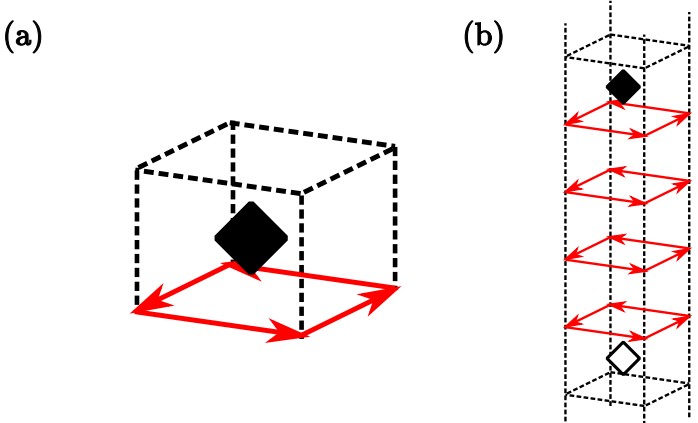

**Figure 2.** The link-variables to realize a static monopole–antimonopole pair on the lattice in spatial $\mathbf{R}^3$. Only the red link-variables take a non-trivial value of $i$. (**a**) The cube including only one all-red plaquette induces a magnetic monopole at its inside on the dual lattice. (**b**) A monopole (black diamond) and an anti-monopole (white diamond) appear at the two terminals of the red plaquette tower.

This monopole and anti-monopole system has also physical magnetic flux around the line segment connecting the monopole pair. In fact, a physical magnetic field is created in the neighboring plaquette $u_{xy}(s)$ of the all-red plaquette in Figure 1. In this idealized system, only the plaquette $u_{xy}(s)$ including one red link takes a non-trivial value as

$$u_{xy}(s) = -i = e^{-i\pi/2} \quad \text{in case with one nontrivial link}, \quad (30)$$

otherwise $u_{\mu\nu}(s) = 1$. Note here that, by gauge transformation, the location of the Dirac string is generally changed, but the physical field strength is never changed.

Figure 3 shows the local chiral condensate $\langle \bar{\chi}\chi(s) \rangle_u$ and the magnetic quantity $\mathcal{F} \equiv \frac{1}{4}F_{\mu\nu}^2 = \frac{1}{2}\vec{H}^2$ for $l = 4$ in the three dimensional space $\mathbf{R}^3$. In this demonstration, the quark mass is taken to be $m = 0.01$ in the lattice unit.

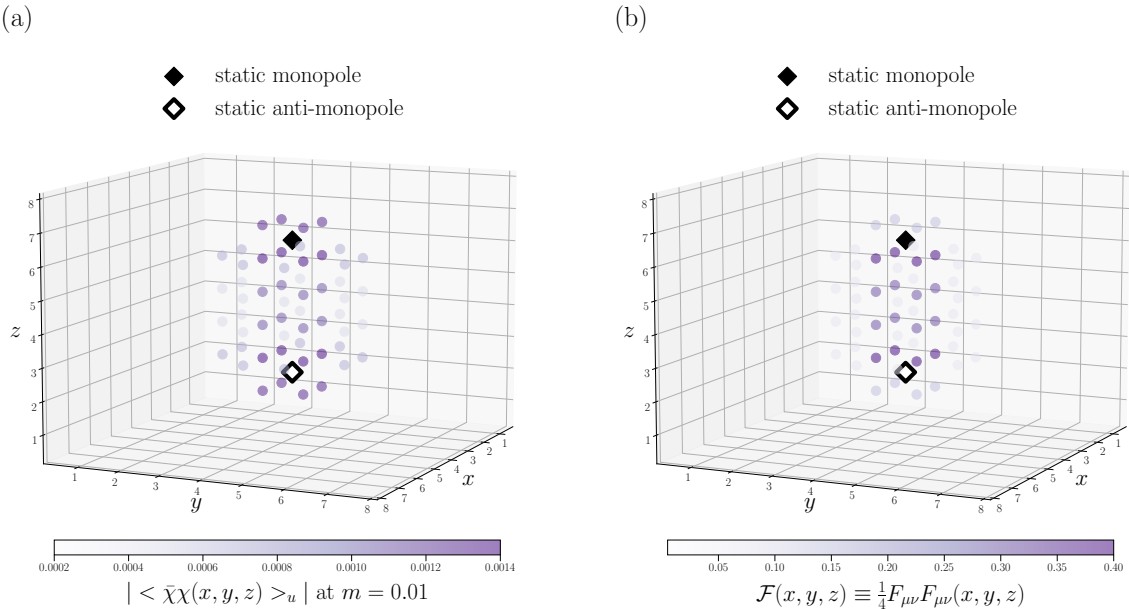

**Figure 3.** An idealized Abelian gauge system of a static monopole and anti-monopole pair with the distance of $l = 4$. In the three dimensional space $\mathbf{R}^3$, the value is visualized with the color graduation for (**a**) the local chiral condensate and (**b**) the magnetic quantity $\mathcal{F}$.

For this idealized static system, there actually appears a magnetic field $\vec{H}$, i.e., non-zero flux of $\mathcal{F} = \frac{1}{2}\vec{H}^2 > 0$, in space between the monopole and the anti-monopole, and the local chiral condensate takes a significant value in the vicinity of the magnetic field. In contrast, one finds $\mathcal{G} = \vec{H} \cdot \vec{E} = 0$ everywhere, since only spatial plaquettes take a non-trivial value and $\vec{E} = \vec{0}$. Thus, in this system, it is likely that the magnetic field stemming from monopoles has the primary correlation with the local chiral condensate.

### 5.2. Static Magnetic Flux System

Next, let us investigate a static magnetic flux system without monopoles. Owing to the spatial periodicity, the special case of $l = L_z$ in the static monopole–antimonopole system has no (anti)monopoles, because of the magnetic-charge cancellation. In this special case of $l = L_z$, there only exists a physical static magnetic flux along $z$-direction.

Figure 4 shows the local chiral condensate $\langle \bar{\chi}\chi(s) \rangle_u$ and the magnetic quantity $\mathcal{F} \equiv \frac{1}{4}F_{\mu\nu}^2 = \frac{1}{2}\vec{H}^2$ for $l = L_z$ in spatial $\mathbf{R}^3$, taking the quark mass of $m = 0.01$ in the lattice unit.

Again, the local chiral condensate takes a significant value in the vicinity of the magnetic field $\vec{H}$, i.e., non-zero flux of $\mathcal{F} = \frac{1}{2}\vec{H}^2 > 0$, even without (anti)monopoles. Note also that this system has $\mathcal{G} = \vec{H} \cdot \vec{E} = 0$ everywhere, because of $\vec{E} = \vec{0}$. Therefore, in this idealized system, we conclude that the magnetic field or $\mathcal{F}$ has the primary correlation with the local chiral condensate.

(a)                                    (b)

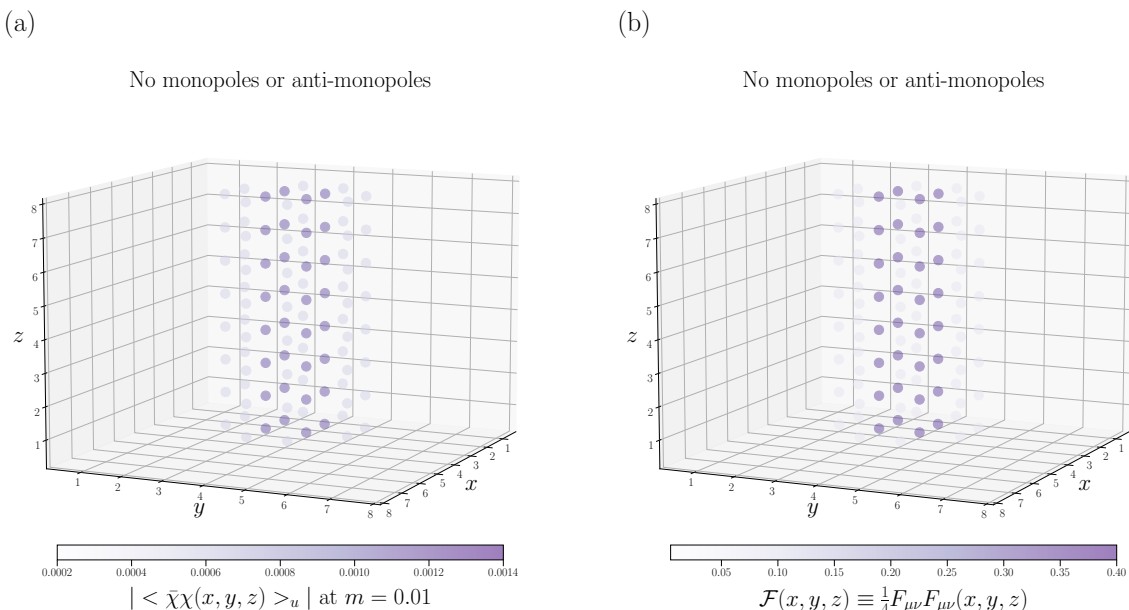

**Figure 4.** An idealized Abelian gauge system of a static magnetic flux without monopoles. In the three dimensional space $\mathbf{R}^3$, the value is visualized with the color graduation for (**a**) the local chiral condensate and (**b**) the magnetic quantity $\mathcal{F}$.

## 6. Lattice QCD Study for Local Chiral Condensate, Monopoles, and Magnetic Fields

In our previous study with lattice QCD, we observed a strong correlation between the local chiral condensate and monopoles in Abelian projected QCD [33]. As a possible reason of this correlation, we conjectured that the strong magnetic field around monopoles is responsible to chiral symmetry breaking in QCD, similarly to the magnetic catalysis [46–49].

In this section, using lattice QCD Monte Carlo simulations, we investigate the relation among the local chiral condensate, monopoles, and magnetic fields in Abelian projected QCD. In this paper, the SU(3) lattice QCD simulation is performed using the standard plaquette action at the quenched level. In each space-time direction, we impose the periodic boundary condition for link variables, and the anti-periodic boundary condition for quarks in order to describe also thermal QCD.

For the numerical Monte Carlo calculation, we basically adopt the lattice parameter of $\beta \equiv 2N_c/g^2 = 6.0$ and the size $V = 24^4$. The lattice spacing $a \simeq 0.1$ fm is obtained from the string tension $\sigma = 0.89$ GeV/fm [37]. Additionally, we adopt $\beta = 6.0$ and $V = 24^3 \times 6$ for the high-temperature deconfined phase at $T \simeq 330$ MeV above the critical temperature.

Using the pseudo-heat-bath algorithm, we generate 100 and 200 gauge configurations for $V = 24^4$ and $24^3 \times 6$, respectively. All the gauge configurations are taken every 500 sweeps after thermalization of 5000 sweeps. MA gauge fixing is performed with the stopping criterion that the deviation $\Delta R_{\mathrm{MA}}/(4V)$ becomes smaller than $10^{-5}$ in 100 iterations. For the calculation of the local chiral condensate, we use the quark propagator of the KS fermion with the quark mass of $m = 0.01, 0.015, 0.02$ in the lattice unit, Here, the quark mass is taken to be finite, since the chiral and continuum limits do not commute for the KS fermion at the quenched level [45]. The jackknife method is used for statistical error estimates.

For each lattice gauge configuration of Abelian projected QCD in the MA gauge, we calculate the local monopole density $\rho_{\mathrm{L}}(s)$, the local chiral condensate, and the Lorentz invariants $\mathcal{F}$ and $\mathcal{G}$, defined in Section 3.

### 6.1. Distribution Similarity between Local Chiral Condensate and Magnetic Variables

To begin with, we pick up a gauge configuration generated in lattice QCD on $V = 24^4$ at $\beta = 6.0$, and investigate correlation between the local chiral condensate and magnetic variables.

Figure 5 shows the local chiral condensate $\langle \bar{\chi}\chi(s) \rangle_u$ with the quark mass of $m = 0.02$, the local monopole density $\rho_L(s)$, and the Lorentz invariants $\mathcal{F}(s)$ and $|\mathcal{G}(s)|$, respectively, as well as the monopole location in the space $\mathbf{R}^3$ at a time slice, for a typical gauge configuration of Abelian projected QCD.

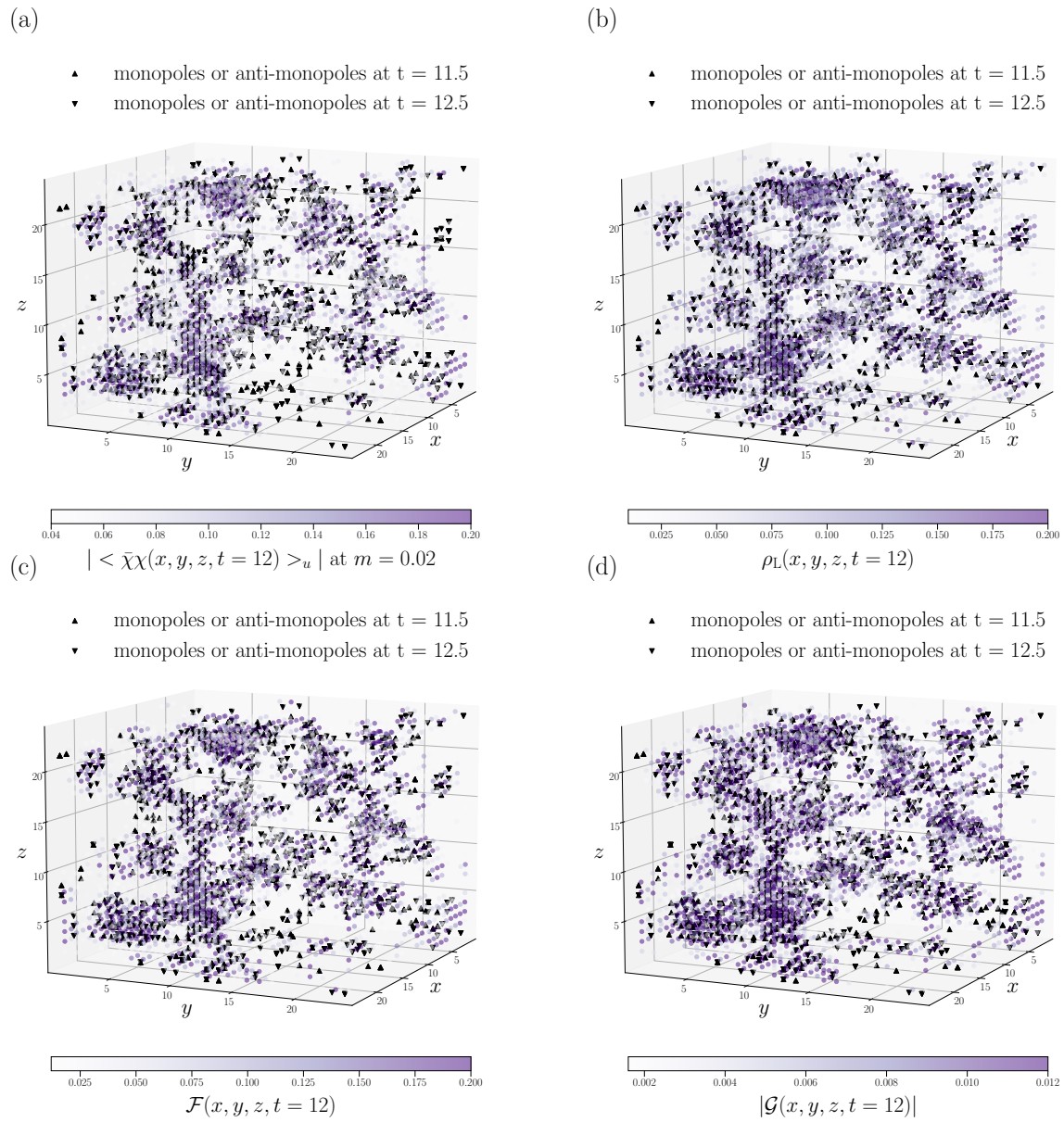

**Figure 5.** Lattice QCD results for (**a**) the local chiral condensate $\langle \bar{\chi}\chi(s) \rangle_u$ with the quark mass of $m = 0.02$, (**b**) the local monopole density $\rho_L(s)$, and the Lorentz invariants (**c**) $\mathcal{F}(s)$ and (**d**) $|\mathcal{G}(s)|$ in spatial $\mathbf{R}^3$ at a time slice, for a typical gauge configuration of Abelian projected QCD. The value is visualized with the color graduation. Monopoles at $t = 11.5$ and 12.5 are plotted with upper and lower triangles, respectively.

From Figure 5a, one finds that the local chiral condensate tends to take a large value near the monopole location [33]. Since monopoles appear on the dual lattice, we show the local monopole density $\rho_L(s)$, as the average on closest dual sites. Of course, $\rho_L(s)$ takes a large value near the monopole. The distribution of the the local monopole density resembles that of the local chiral condensate, as was pointed out in Ref. [33]. Figure 5c,d show the Lorentz invariants $\mathcal{F}$ and $\mathcal{G}$, respectively. As a new result in this paper, we find that the distributions of $\mathcal{F}$ and $\mathcal{G}$ also resemble that of the local chiral condensate.

The close relation of monopoles with $\mathcal{F}$ and $\mathcal{G}$ might be understood, since the field strength tensor relates to monopoles as $\partial_\mu \tilde{F}^i_{\mu\nu} = k^i_\nu$. Roughly speaking, the monopole can be a kind of source of $\mathcal{F}$ and $\mathcal{G}$. In contrast, their similarity with the local chiral condensate is fairly non-trivial.

In any case, we find clear correlation of distribution similarity among the local chiral condensate, the local monopole density, and the Lorentz invariants $\mathcal{F}$ and $\mathcal{G}$ in Abelian projected QCD in the MA gauge.

### 6.2. Correlation Coefficients between Local Chiral Condensate and Magnetic Variables

In this subsection, we quantify the similarity between the local chiral condensate $\langle \bar{\chi}\chi(s)\rangle_u$ and magnetic variables, i.e., $\rho_{\rm L}(s)$, $\mathcal{F}(s)$ and $\mathcal{G}(s)$, defined in Section 3. To this end, we use all the generated 100 gauge configurations in lattice QCD on $V = 24^4$ at $\beta = 6.0$, and calculate the local chiral condensate at $2^4$ distant space-time points for each gauge configuration, resulting 1600 data points at each quark mass.

Figure 6 shows the scatter plot between the local chiral condensate $\langle \bar{\chi}\chi(s)\rangle_u$ and magnetic variables, i.e., the local monopole density $\rho_{\rm L}(s)$, Lorentz invariants $\mathcal{F}(s)$ and $|\mathcal{G}(s)|$, respectively, using 100 gauge configurations of Abelian projected QCD in the MA gauge, with the quark mass of $m = 0.01, 0.015, 0.02$ in the lattice unit. In Figure 6, positive correlation is qualitatively found between the local chiral condensate and the magnetic variables, $\rho_{\rm L}$, $\mathcal{F}$ and $|\mathcal{G}|$, respectively.

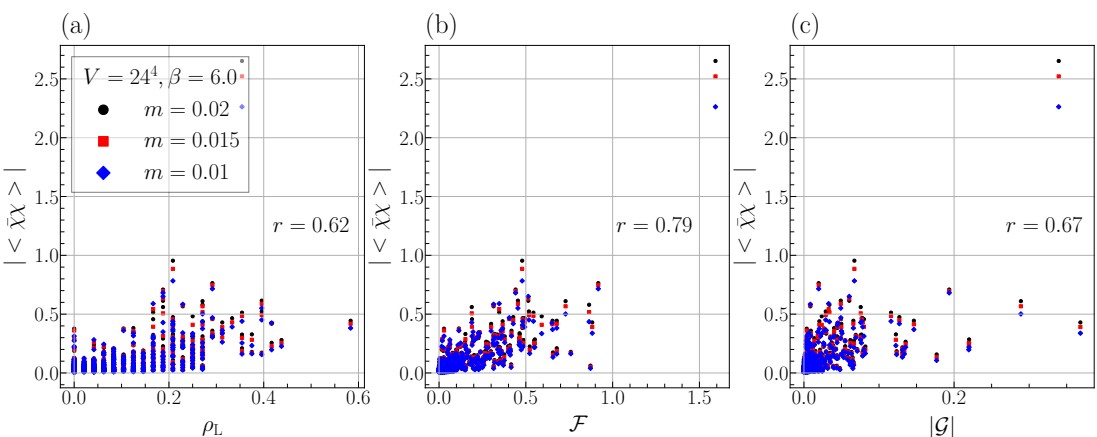

**Figure 6.** The scatter plot between the local chiral condensate $\langle \bar{\chi}\chi(s)\rangle_u$ and (**a**) the local monopole density $\rho_{\rm L}(s)$, (**b**) $\mathcal{F}(s)$ and (**c**) $|\mathcal{G}(s)|$, using 100 Abelianized gauge configurations in SU(3) lattice QCD with $\beta = 6.0$ and $V = 24^4$ at each quark mass $m$.

Next, we consider a quantitative analysis using correlation coefficients between the local chiral condensate and the magnetic variables, as a statistical indicator of correlation. In general, for arbitrary two statistical ensembles $\{A_i\}$ and $\{B_i\}$, their correlation coefficient $r$ is defined as

$$r \equiv \frac{\langle (A - \langle A \rangle)(B - \langle B \rangle)\rangle}{\sigma_A \sigma_B}, \tag{31}$$

using the average notation $\langle\ \rangle$ and the standard deviation $\sigma_A \equiv \sqrt{\langle (A - \langle A \rangle)^2 \rangle}$ and $\sigma_B \equiv \sqrt{\langle (B - \langle B \rangle)^2 \rangle}$. Here, $r = 1$ means perfect positive linear correlation, and $r \gtrsim 0.7$ indicates strong positive linear correlation.

We measure correlation coefficients between the local chiral condensate $|\langle \bar{\chi}\chi(s)\rangle_u|$ and three magnetic variables, $\rho_{\rm L}(s)^\alpha$, $\mathcal{F}(s)^\alpha$ and $|\mathcal{G}(s)|^\alpha$, at various exponent $\alpha$, using 100 gauge configurations of Abelian projected QCD in SU(3) lattice QCD at $\beta = 6.0$ on $V = 24^4$, for the quark mass $m = 0.01$ in the lattice unit. Table 1 shows the result for the correlation coefficients.

**Table 1.** Correlation coefficients between the local chiral condensate $|\langle\bar{\chi}\chi(s)\rangle_u|$ and three magnetic variables, $\rho_L(s)^\alpha$, $\mathcal{F}(s)^\alpha$, and $|\mathcal{G}(s)|^\alpha$ at various $\alpha$, using 100 gauge configurations of Abelian projected QCD in SU(3) lattice QCD at $\beta = 6.0$ on $V = 24^4$, for the quark mass $m = 0.01$ in the lattice unit.

| Lattice | $\alpha$ | $\rho_L^\alpha$ | $\mathcal{F}^\alpha$ | $|\mathcal{G}|^\alpha$ |
|---------|----------|-----------------|----------------------|------------------------|
| $V = 24^4$, $\beta = 6.0$ | 0.25 | 0.47 | 0.63 | 0.60 |
| | 0.5 | 0.55 | 0.71 | 0.67 |
| | 1 | 0.62 | 0.79 | 0.67 |
| | 1.5 | 0.63 | 0.81 | 0.60 |
| | 2 | 0.60 | 0.80 | 0.55 |

Quantitatively, the magnetic quantity $\mathcal{F}$ has the strongest correlation with the chiral condensate rather than $\rho_L$ and $\mathcal{G}$. As a conclusion of this paper, we find a strong positive correlation of $r \simeq 0.8$ between the local chiral condensate $|\langle\bar{\chi}\chi(s)\rangle_u|$ and the magnetic quantity $\mathcal{F}(s)$ in the confined vacuum of Abelian projected QCD.

*6.3. High-Temperature Deconfined Phase*

Finally, we also investigate a high-temperature deconfined phase in lattice QCD on $V = 24^3 \times 6$ at $\beta = 6.0$, where the temperature is $T \simeq 330$ MeV above the critical temperature. We generate 200 gauge configurations, and calculate the local chiral condensate at $2^3$ distant space points at a time slice for each gauge configuration, resulting 1600 data points at each quark mass.

Figure 7 shows the scatter plot between the local chiral condensate $|\langle\bar{\chi}\chi(s)\rangle_u|$ and magnetic variables, i.e., the local monopole density $\rho_L(s)$, and Lorentz invariants $\mathcal{F}(s)$ and $|\mathcal{G}(s)|$, respectively, using 200 gauge configurations of Abelian projected QCD in the MA gauge, with the quark mass of $m = 0.01, 0.015, 0.02$ in the lattice unit.

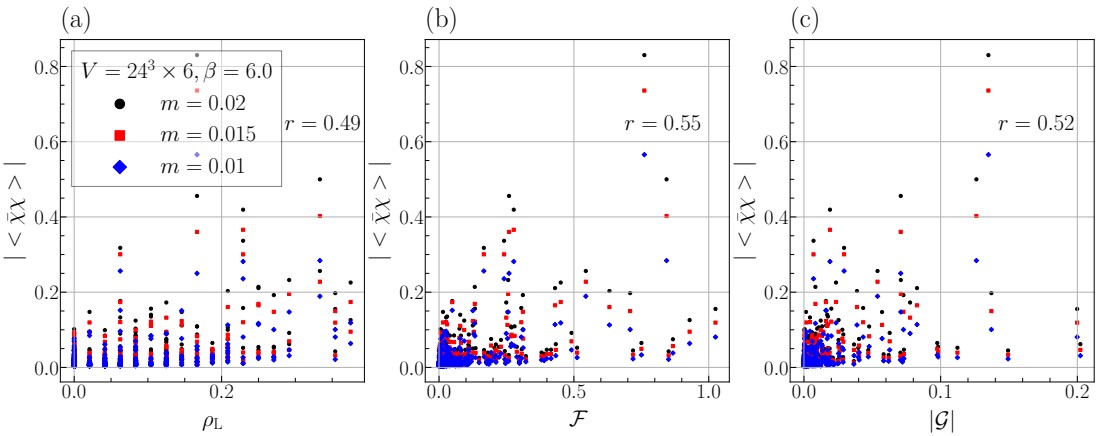

**Figure 7.** Result of the high-temperature deconfined phase for the scatter plot between the local chiral condensate $\langle\bar{\chi}\chi(s)\rangle_u$ and (**a**) $\rho_L(s)$, (**b**) $\mathcal{F}(s)$ and (**c**) $|\mathcal{G}(s)|$, using 200 Abelianized gauge configurations in SU(3) lattice QCD with $\beta = 6.0$ and $V = 24^3 \times 6$ at each quark mass $m$.

We show in Table 2 correlation coefficients between the local chiral condensate $|\langle\bar{\chi}\chi(s)\rangle_u|$ and three magnetic variables, $\rho_L(s)^\alpha$, $\mathcal{F}(s)^\alpha$ and $|\mathcal{G}(s)|^\alpha$, at various exponent $\alpha$, using 200 gauge configurations of Abelian projected QCD in SU(3) lattice QCD at $\beta = 6.0$ on $V = 24^3 \times 6$, for the quark mass $m = 0.01$ in the lattice unit.

From Figure 7 and Table 2, all the correlations between the local chiral condensate and the three magnetic variables, $\rho_L$, $\mathcal{F}$ and $\mathcal{G}$, become weaker in the deconfined phase, where the chiral condensate itself goes to zero in the chiral limit.

**Table 2.** Correlation coefficients in the deconfined phase between the local chiral condensate $|\langle\bar{\chi}\chi(s)\rangle_u|$ and three magnetic variables, $\rho_L(s)^\alpha$, $\mathcal{F}(s)^\alpha$ and $|\mathcal{G}(s)|^\alpha$ at various $\alpha$ in Abelian projected QCD of SU(3) lattice QCD at $\beta = 6.0$ on $V = 24^3 \times 6$ for $m = 0.01$.

| Lattice | $\alpha$ | $\rho_L^\alpha$ | $\mathcal{F}^\alpha$ | $|\mathcal{G}|^\alpha$ |
|---|---|---|---|---|
| $V = 24^3 \times 6$, $\beta = 6.0$ | 0.25 | 0.37 | 0.49 | 0.49 |
| | 0.5 | 0.43 | 0.55 | 0.55 |
| | 1 | 0.49 | 0.55 | 0.52 |
| | 1.5 | 0.50 | 0.51 | 0.45 |
| | 2 | 0.49 | 0.46 | 0.38 |

## 7. Summary and Conclusions

We have studied the relation among the local chiral condensate, monopoles, and magnetic fields, using the lattice gauge theory, as a continuation of Ref. [33].

First, we have created idealized Abelian gauge systems of (1) a static monopole–antimonopole pair, and (2) a magnetic flux without monopoles, on a four-dimensional Euclidean lattice. In these systems, we have calculated the local chiral condensate on quasi-massless fermions coupled to the Abelian gauge field, and have found that the chiral condensate is localized in the vicinity of the magnetic field.

Second, performing SU(3) lattice QCD Monte Carlo simulations, we have investigated Abelian projected QCD in the maximally Abelian gauge, and have found clear correlation of distribution similarity among the local chiral condensate, color monopoles, and color magnetic fields in the Abelianized gauge configuration.

As a statistical indicator, we have measured the correlation coefficient $r$, and have found a strong positive correlation of $r \simeq 0.8$ between the local chiral condensate and the Euclidean color-magnetic quantity $\mathcal{F}$.

We have also examined the local correlation in the deconfined phase of thermal QCD, and have found that the correlation between the local chiral condensate and magnetic variables becomes weaker.

Thus, in this paper, we have observed a strong correlation between the local chiral condensate and magnetic fields in both idealized Abelian gauge systems and Abelian projected QCD. From these results, we conjecture that the chiral condensate is locally enhanced by the strong color-magnetic field around the monopoles in Abelian projected QCD, like magnetic catalysis.

Note, however, that this correlation does not necessarily mean that chiral symmetry breaking is caused by the non-uniform magnetic field. To realize spontaneous chiral-symmetry breaking, as was discussed in Section 4.2, we need some zero mode in the denominator of $I$ in the chiral limit. In the context of the dual superconductor picture, this might be realized by condensation of monopoles, as was suggested in the dual Ginzburg-Landau theory [28].

To conclude, once chiral symmetry is spontaneously broken, the local chiral condensate is expected to have a strong correlation with the color magnetic field.

**Author Contributions:** Conceptualization, H.S.; methodology, H.S.; software, H.O.; validation, H.O.; formal analysis, H.S. and H.O.; investigation, H.O.; resources, H.O.; data curation, H.O.; writing—original draft preparation, H.S. and H.O.; writing—review and editing, H.S. and H.O.; visualization, H.O.; supervision, H.S.; project administration, H.S.; funding acquisition, H.S. and H.O. All authors have read and agreed to the published version of the manuscript.

**Funding:** H.S. was supported in part by the Grants-in-Aid for Scientific Research [19K03869] from Japan Society for the Promotion of Science. H.O. was supported by a Grant-in-Aid for JSPS Fellows (Grant No. 21J20089).

**Data Availability Statement:** The basic data of this study are available from the authors.

**Acknowledgments:** Most of numerical calculations have been performed on OCTOPUS, at the Cybermedia Center, Osaka University. We have used PETSc to solve linear equations for the Dirac operator [50–52].

**Conflicts of Interest:** The authors declare no conflict of interest.

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
