# Peer review of "Local Correlation among the Chiral Condensate, Monopoles, and Color Magnetic Fields in Abelian Projected QCD"

_universe, doi:10.3390/universe7090318_

Round 1

Reviewer 1 Report

The authors present a study about the correlation between the Chiral condensate, field strengths, and monopole configurations in this work. Even though they advertise results for the SU(3) system, these are not new. All the new results are for abelian systems. 

The paper is scientifically sound and the results are of some interest. The problem is that the most relevant conclusions, regarding the SU(3) case, were already presented in another publication by the same authors, [Phys. Rev. D 2021, 103, 054505]. The present paper is presented as a continuation of that other publication. The new content is a discussion of the analytic expression of the chiral condensate as a function of background field strengths in the abelian theory, the lattice study of a configuration of a pair of static monopole and anti-monopole in the abelian theory, and also a study of magnetic flux without monopoles. There is nothing fundamentally new in these additions, the analytic expression is trivial and the study of the pair of monopoles and flux are special cases of the results already presented in the other publication.

Nevertheless, these results may be of some pedagogical interest to the community because they help make the case for the claimed correlation between the chiral condensate and field strength configurations. That is why I think it deserves to be published. But the authors should first be more clear about what is the new contribution here and what was already discussed in the other publication. They do comment about it in the text, but it is not enough. Besides, some English typos should be corrected.

Author Response

----------------------------------------------------------------------
 Summary of Changes
----------------------------------------------------------------------

1. Following the referee's suggestion to measure the correlation coefficient, 
   we have newly performed lattice QCD calculations and have measured 
   the correlation coefficient between the local chiral condensate and magnetic variables. 
   This result is summarized in a new subsection of Section 6.2 in the revised version.

2. Following the referee's suggestion, we have clarified that this paper is a continuation 
   of our previous paper, and that the new point is the correlation with magnetic fields.

3. Considering the referee's comment, we have added a brief explanation for 
    the chiral condensate in the free case with F=0. 

4. Following the referee's suggestion, we have adopted a modest estimate on the 
    chiral symmetry breaking contribution, and have introduced evidence of monopole
    condensation without gauge fixing, and have added a caution that the chiral and 
    continuum limits do not commute in the KS fermion at the quenched level. 

5. We have newly added the similar statistical analysis for the deconfined phase 
   in lattice QCD using 200 gauge configurations.
   This result is summarized in a new subsection of Section 6.3 in the revised version.

----------------------------------------------------------------------
 Answer to the comments of Referees
----------------------------------------------------------------------

We thank the referees for reviewing our manuscript entitled 
"Local correlation among the chiral condensate, monopoles, and color magnetic fields 
in Abelian projected QCD" authored by H. Suganuma and H. Ohata.
We sincerely thank also their valuable comments and suggestions.

Following the referee's suggestion, we have revised our manuscript as faithful as possible.
The following is the answer to the referee report.

Answer to Report of Referee 1:
============================================

> The problem is that the most relevant conclusions, regarding the SU(3) case, were 
> already presented in another publication by the same authors, [Phys. Rev. D 2021, 103, 
> 054505]. The present paper is presented as a continuation of that other publication.  
> ... But the authors should first be more clear about what is the new contribution here 
> and what was already discussed in the other publication. They do comment about it in the 
> text, but it is not enough. 

We agree with the referee. The present study is a continuation of our previous work of 
Ref.[33] (Phys. Rev. D103 (2021) 054505), which was shortly noted in the previous version.

Following the referee's suggestion, we have explicitly clarified that this paper is 
a continuation of our previous paper in the introduction and in the summary.
The new point of this paper is the correlation with magnetic fields.
Then, we have clarified the new point in the introduction in the revised version. 
For instance, we have added the following sentences in the last paragraph in Section 1:

  In this paper, as a continuation of Ref.[33], we proceed lattice works for the relation 
  between chiral symmetry breaking and color magnetic objects including monopoles.
  In particular, as a new point of this paper, we quantitatively study correlation of 
  the local chiral condensate with color magnetic fields using the lattice gauge theory.

Also, we have clarified the review part by the explicit use of "we review..." or "we prepare..."
in the revised version. 

> Besides, some English typos should be corrected.

We have corrected typos throughout the manuscript as possible.

We thank the referee again for the encouraging comment that our results may be of some 
pedagogical interest to the community. 

Answer to Report of Referee 2:
============================================

> Most importantly, the central claim of the paper relies on Fig. 1, which is a single time 
> slice of a single configuration in their ensemble. Furthermore, the recognition of 
> similarity is left to the pattern matching algorithm of the human eye, which can be 
> deceiving. I would strongly suggest that the authors find some qualitative criterion - 
> like a correlation coefficient - that they measure for every thermalized config in 
> their ensemble to show that the similarity is statistically relevant.

We completely agree with the referee.
Following the referee's suggestion, we have newly performed lattice QCD calculations 
and have measured the correlation coefficient between the local chiral condensate 
and magnetic variables as a statistical indicator of the correlation or the similarity, 
using 100 gauge configurations.
In the revised version, we have added this result in a new subsection of Section 6.2.

Owing to the statistical measurement with the correlation coefficients, 
the present result becomes clearer. 
We sincerely thank the referee for the valuable comment.

Also, we have added a new subsection of Section 6.3 for the similar statistical analysis 
for the deconfined phase in lattice QCD using 200 gauge configurations.

> I do not quite understand the argument following (20). I agree that the operator 
> (D^2+m^2)^2 is positive definite for any finite (real) m. However, as the limit m->0 is 
> considered, D^4 itself is only positive semidefinite. In particular, in the presence of 
> zero modes, I do not understand how the argument presented holds. Of course there 
> are no topological zero modes in 4D U(1), but for the free operator in particular that 
> is mentioned by the authors, zero modes are present and it seems to me that
> I diverges as 1/m.

Considering the referee's comment, we have added a brief explanation for 
the chiral condensate in the free case with F=0 around Eq.(22) in Section 4.2. 
Our explanation in the previous version might be unclear.

> Minor points:
> The statement in the introduction about chiral symmetry breaking contributing 99% to 
> the non-dark matter content of the universe is a bit too much. It probably comes from 
> a naive comparison of the MSbar masses of u and d compared to the nucleon mass.
> Realistic estimates need to consider nucleon structure like e.g. Phys.Rev.D 52 (1995) 
> 271-281.

We agree with the referee on our naive evaluation.
Following the referee's suggestion, we have adopted a modest estimation on the 
chiral symmetry breaking contribution to the non-dark matter content of our Universe 
in the introduction in the revised version. 

> In the introduction the authors convey the impression, that support for the dual 
> superconductor picture comes solely from the MAG/Abelian projection. This seems 
> to ignore that there is evidence for the condensation color-magnetic monopoles in 
> nonabelian gauge theories without gauge fixing, e.g. Phys.Rev.D 63 (2001) 034506.

Following the referee's suggestion, we have referred the work on evidence of 
monopole condensation without gauge fixing, with mentioning that MA gauge fixing 
is a concrete way, in the introduction of the revised version as follows:  

 Note however that, even without gauge fixing, there is an evidence of monopole  
 condensation in nonabelian gauge theories [27], and therefore it might be possible 
 to define infrared-relevant monopoles in QCD and to construct the dual 
 superconductor system in more general manner. 
 In fact, MA gauge fixing gives a concrete way to extract infrared-relevant 
 Abelian gauge manifold and monopoles from QCD.

> It is well known that for staggered fermions the chiral and continuum limits do not 
> commute. No such limit is performed in the paper, but it is frequently referred to 
> in the analytic section. Also the actual quark mass used in the numerical part is 
> probably large enough not to be substantially affected by this. Still it would be nice 
> to have this potential difficulty mentioned as a matter of principle.

We agree with the referee on the caution. 
Following the referee's suggestion, we have added a caution that the chiral and 
continuum limits do not commute in the KS fermion at the quenched level 
with citing Ref.[45] (C. Bernard, Phys Rev D71 (2005) 094020.) in Section 4.1 and 
in the fourth paragraph of Section 6. 

Owing to the major revision following the referees' suggestions including 
the new lattice analysis, we think that the revised manuscript becomes much more 
useful and readable.

We hope that the current version meets the requirement for publication in 
Special Issue of Universe.

H. Suganuma and H. Ohata

Reviewer 2 Report

In the manuscript "Local correlation among the chiral condensate, monopoles, and color magnetic fields in Abelian projected QCD", the authors present some interesting correlations between the quantities mentioned in the title. The paper contains a numerical simulation in quenched QCD and the investigation of an artificial monopole-antiminopoloe system. While I think that the topic and the findings are in principle interesting enough to warrant publication, I find the analysis methods a bit lacking and have some questions regarding their analytic results and more generic statements. I therefore think the manuscript is in principle suitable for publication, but needs some revision.

Details:

.) Most importantly, the central claim of the paper relies on Fig. 1, which is a single time slice of a single configuration in their ensemble. Furthermore, the recognition of similarity is left to the pattern matching algorithm of the human eye, which can be deceiving. I would strongly suggest that the authors find some qualitative criterion - like a correlation coefficient - that they measure for every thermalized config in their ensemble to show that the similarity is statistically relevant.

.) I do not quite understand the argument following (20). I agree that the operator (D^2+m^2)^2 is positive definite for any finite (real) m. However, as the limit m->0 is considered, D^4 itself is only positive semidefinite. In particular, in the presence of  zero modes, I do not understand how the argument presented holds. Of course there are no topological zero modes in 4D U(1), but for the free operator in particular that is mentioned by the authors, zero modes are present and it seems to me that I diverges as 1/m.

Minor points:

.) The statement in the introduction about chiral symmetry breaking contributing 99% to the non-dark matter content of the universe is a bit too much. It probably comes from a naive comparison of the MSbar masses of u and d compared to the nucleon mass. Realistic estimates need to consider nucleon structure like e.g. Phys.Rev.D 52 (1995) 271-281.

In the introduction the authors convey the impression, that support for the dual superconductor picture comes solely from the MAG/Abelian projection. This seems to ignore that there is evidence for the condensation color-magnetic monopoles in nonabelian gauge theories without gauge fixing, e.g.  Phys.Rev.D 63 (2001) 034506.

It is well known that for staggered fermions the chiral and continuum limits do not commute. No such limit is performed in the paper, but it is frequently referred to in the analytic section. Also the actual quark mass used in the numerical part is probably large enough not to be substantially affected by this. Still it would be nice to have this potential difficulty mentioned as a matter of principle.

Author Response

(The authors gave the same response as above.)
